# New estimations of child marriage: Evidence from 98 low- and middle-income countries

**Mengjia Liang**[1]*, **Sandile Simelane**[1], **Satvika Chalasani**[2], **Rachel Snow**[1]

**1** Population and Development Branch, United Nations Population Fund, New York City, New York, United States of America, **2** Sexual and Reproductive Health Branch, United Nations Population Fund, New York City, New York, United States of America

* liang@unfpa.org

## Abstract

The Sustainable Development Goals include a target on eliminating child marriage, a human rights abuse. Yet, the indicator used in the SDG framework is a summary statistic and does not provide a full picture of the incidence of marriage at different ages. This paper aims to address this limitation by providing an alternative method of measuring child marriage. The paper reviews recent data on nuptiality and captures evidence of changes in the proportion married and in the age at marriage, in 98 low- and middle-income countries (LMICs). Using data collected from nationally representative Demographic and Health Surveys and Multiple Indicator Cluster Surveys, survival analysis is applied to estimate (a) age-specific marriage hazard rates among girls before age 18; and (b) the number of girls that were married before age 18 in 2020. Results show that the vast majority of girls remain unmarried until age 10. Child marriage rates increase gradually until age 14 and accelerate significantly thereafter at ages 15–17. By accounting for both single-year-age-specific child marriage hazard rates and the age structure of the population with a survival analysis approach, lower estimates in countries with a rapid decrease in child marriage and higher estimates in countries with constant or slightly rising child marriage rates relative to the direct approach are obtained.

## Background

Marriage is typically studied among demographers as an established precursor to fertility or in the interest of its possible implications for fertility. However, child marriage, defined as a formal marriage or informal union in which one or both of the parties are under 18 years of age, has diverse implications for the lives of young girls who marry early as they are likely to be at greater risks of school drop-out and illiteracy, of poorer engagement and earnings in the work force, of less decision-making power, and of less control over valuable household assets [1, 2]; child marriage also impacts young brides' early and long-term empowerment within marriage [1, 3]. In many countries, child marriage is one of the most common reasons for girls to drop-out school [4, 5]. Sekine and colleagues found that married girls aged 15–17 in Nepal were 10 times more likely to drop out of school compared to their unmarried peers [6]. Hence, for a

**Data Availability Statement:** All data is taken from the publicly available Demographic and Health Surveys (http://dhsprogram.com) and Multiple Indicator Cluster Surveys (http://mics.unicef.org).

**Funding:** The author(s) received no specific funding for this work.

**Competing interests:** The authors have declared that no competing interests exist.

girl, marriage can mean the end of her education, can derail her chances of a vocation or career, and can steal from her the chance to make foundational life choices.

Child marriage is significantly associated with high fertility and poor pregnancy spacing, including a repeat childbirth in less than 24 months, multiple unwanted pregnancies, pregnancy termination, and sterilization [7–9]. Since child marriage so often leads to early childbearing or unintended pregnancy, it also significantly impacts the health of both young mothers and their children [7, 10–12]. This is particularly salient as pregnancy complications, such as haemorrhage, sepsis, obstructed labour, and complications from unsafe abortions, are the leading cause of death among 15–19-year-old girls [13]. Girls who are married are also more exposed to sexually transmitted infections, including HIV [14, 15]. Significant association has also been identified between maternal child marriage and poor health outcomes of both infants and children, including greater risk of stunting and underweight [12, 16].

Child marriage is also a known risk factor for intimate partner violence [17–21]. Using standardized data in 34 countries, Kidman (2016) showed that compared to women who married as adults, young women aged 20–24 years who married as children were at greater risk of physical and/or sexual intimate partner violence in the past 12 months [odds ratio (OR) 1.41 (1.30–1.52) for marriage before 15, and 1.42 (1.35–1.50) for marriage at 15–17] [22].

Child marriage is addressed under various international treaty bodies, the Convention on the Elimination of all forms of Discrimination Against Women (CEDAW) and the Convention on the Rights of the Child (CRC), as well as in landmark international agreements such as the International Conference of Population and Development Program of Action (ICPD-POA) and the Beijing Declaration and Platform for Action. World leaders made a collective commitment to eliminate child marriage and included it as one of the targets for the sustainable development agenda 2030. Goal 5, "achieve gender equality and empower all women and girls" of the Sustainable Development Goal (SDG) framework includes Target 5.3, which is "eliminate all harmful practices, such as child, early and forced marriage and female genital mutilation". It is worth noting that SDG Goal 5, and its associated targets, are an extension of a corresponding Millennium Development Goal–MDG Goal 3 –"promote gender equality and empower women".

The indicator used to track progress towards eliminating child marriage in the SDG global framework is SDG indicator 5.3.1 "Proportion of women aged 20–24 years who were married or in a union before age 15 and before age 18". While widely used and analyzed in population and development monitoring tools and studies, this indicator is only a summary statistic of the prevalence of child marriage and does not provide a full picture of the incidence of marriage and informal unions among girls at different ages of childhood. Moreover, the SDG summary measure focuses on the 20–24 age group, which is conditional on girls living until they are 20–24 years old. This selection bias is likely to result in an underestimation since girls who marry at a young age are less likely to live to the age of 20 and are thus excluded from the estimates, possibly due to the adverse health outcomes noted previously. This paper aims to reduce the limitations, provide an alternative measurement of child marriage, and further explore the dynamics of marriage among girls using survival analysis—in demography also called event history analysis.

## Measurement of child marriage

Before comparing different sources of marriage data, a number of common conceptual and measurement issues need to be discussed, in particular, the definition of marriage. Both population and housing censuses (Hereafter, simply referred to as "censuses".) and surveys tend to keep the description of marital status simple assuming respondents are reasonably certain

about their current marital status. But in reality, people's understanding of "marriage" and "being married" is varied across societies. The World Fertility Surveys (WFS) and Demographic and Health Surveys (DHS) have focused on one specific aspect of the marital state–exposure to sexual intercourse–and consider cohabitation as the single most critical criterion defining a marriage. Hence, the differences in the definition of marriage between censuses and surveys may yield different tabulations of a country's population by current marital status categories even if the survey and the census are conducted at the same time. One likely explanation of the difference is that the criterion of cohabitation (part of a union according to DHS definition) is not perfectly aligned with those who would have defined themselves as married to a census enumerator [23].

In addition to the haziness of the marriage definition, data collection on age at first marriage can be more challenging than on current marital status, as some women may not recall the date of their first marriage or not want to report the age of first marriage. Missing information includes age, month and year of first marriage. If any of the components of age, month, and year are absent or inconsistent, the answer is deemed incomplete [24]. According to a DHS methodological report assessing quality of age at date reports, using DHS surveys conducted since 2000 in 67 countries, 32 surveys were identified with at least 50% incompleteness in women's reported age at first marriage [24]. Women's reports of their age and date at first marriage have higher levels of incompleteness than their age and birthdate, by an average of approximately 31%, owing mostly to the omission of a month [24]. The accuracy of age at first marriage reporting may also be affected by recall bias or social desirability bias. Women aged 15–19 were less likely to report marriages before the age of 15 than women from the same birth cohort were interviewed five years later at ages 20–24, according to a study based on DHS data from nine African and Latin American countries [25]. The exact impact of the biases on child marriages estimates depends on the specific contexts including the underlying social and cultural differences across the societies. Fear of penalties may lead to overstatement of age at marriage in many countries because the legal age of marriage is older than the cultural norm [26]. Women in Bangladesh, on the other hand, have been shown to deliberately lower their age of marriage, probably because early marriage is socially desirable for women there [25, 27]. Nevertheless, an analysis of longitudinal DHS data from three rounds of a population-based cohort in eastern Zimbabwe found that, while many reports of age at marriage were deemed problematic, their inclusion did not result in artificial generation or suppression of trends [28]. The assessment of age reporting of the top-five countries with the largest number of child brides (India, Bangladesh, Nigeria, Pakistan and Ethiopia), accounting for more than 50% of the global number of child brides also shows data of acceptable quality with some age heaping in ages 10, 12 and 18.

The civil registration and vital statistics (CRVS) systems should ideally provide real time records of vital events, notably, live births, deaths, and marriages and divorces [29]. In reality, most LMICs have hitherto not achieved fully functional and up to date CRVS systems. As a result, information on the levels, patterns and trends of demographic events is primarily derived from household-based enquiries, i.e. censuses and household surveys, often by using indirect estimation techniques. Data on nuptiality patterns is even more limited from CRVS systems because marriages and divorces are not accorded the same level of importance as births and deaths, and assessment of completeness of marriage and divorce registration is often unavailable [30]. This is itself a cause for action, because marriage registration is critically important for safeguarding rights of individuals, particularly women and girls. In the case of child marriage, the event may not be recorded in CRVS systems in countries where marriage under age 18 is considered illegal.

Currently, despite incomplete responses, nationally representative and standardized household surveys such as the DHS and the Multiple Indicator Cluster Surveys (MICS) are the best

sources of internationally comparable data on nuptiality patterns across countries and for the same country over time. This is in contrast to censuses wherein different countries use varied questions and different reference population groups when collecting information on marital behavior. For example, in a total of 144 countries that conducted a population census in the 2010 round of censuses (2005–2014) [31, 32], collected information on marital status, and had a census questionnaire accessible online for analysis, we found eight different reference populations for marital status questions. These include: all members of the household (58 countries); people aged 10 years and above (17 countries), 12 years and above (26 countries), 13 years and above (2 countries), 14 years and above (2 countries), 15 years and above (33 countries), 16 years and above (5 countries), and 18 years and above (1 country).

The lack of standardization of questions on marital behavior in censuses also reflects the fact that not all countries ask about the age at first marriage, and different countries capture different categories of marital status or define categories differently. For example, the category "living together as married/cohabiting" is not captured in all countries and the category "single" is defined differently across countries. In some countries "single" denotes people who have never married or have never been in union, while in other countries it is used to designate people who are single at the time of enumeration, regardless of whether or not they have been married in the past. Among the 144 countries in the 2010 round of censuses referred to above, less than half (43%) explicitly captured information on informal unions (cohabitation) and even fewer countries (18%) collected information on the age at first marriage among respondents who were ever married or ever in union at the time of enumeration.

The DHS and MICS address the limitations of the censuses and provide internationally comparable data on marital behavior because they use a standardized set of questions which are asked of the same reference population. The women's modules of the DHS and MICS surveys specifically ask respondents the following questions in Table 1. The set of questions is consistent to a great extent across multiples rounds of DHS and MICS surveys covered in this study.

Consequently, this paper examines the age-specific rates of early marriage among women in LMICs using the DHS and MICS data.

## Methods

Although survival analysis techniques were designed for longitudinal data [33], their application to cross-sectional data is not uncommon in social sciences [34]. In mortality studies, for

**Table 1. Standard questions used for child marriage measurement in DHS and MICS women's questionnaires.**

| | DHS Phase 5–8 | | MICS Round 4–6 | |
|---|---|---|---|---|
| Q1 | Are you currently married or living together with a man as if married? The pre-coded responses are: (1) yes, currently married; (2) yes, living with a man; and (3) no, not in union; | If response = 1 or 2, move to Q4; if = 3, move to Q2 | Are you currently married or living together with someone as if married? The pre-coded responses are: (1) yes, currently married; (2) yes, living with a partner; and (3) no, not in union; | If response = 1 or 2, move to Q4; if = 3, move to Q2 |
| Q2 | Have you ever been married or lived together with a man as if married? The pre-coded responses are: (1) yes, formerly married; (2) yes, lived with a man; and (3) no; | If response = 1 or 2, move to Q3; if = 3, code as "never married/ single" | Have you ever been married or lived together with someone as if married? The pre-coded responses are: (1) yes, formerly married; (2) yes, formerly lived with a partner; and (3) no; | If response = 1 or 2, move to Q3; if = 3, code as "never married/ single" |
| Q3 | What is your marital status now: are you widowed, divorced or separated? | Move to Q4 | What is your marital status now: are you widowed, divorced or separated? | Move to Q4 |
| Q4 | How old were you when you first started living with him (first husband/partner)? | Ask to all ever-married and ever-in-union respondents | How old were you when you first started living with your (first) husband/partner? | Ask to all ever-married and ever-in-union respondents |

example, demographers differentiate between period (cross-sectional) and cohort (longitudinal) life tables on the basis of the type of data that are used to generate the life table. The computation of the period life tables is based on a synthetic cohort conceptualization, i.e. the life tables show what would happen to a real cohort if it were subjected, for all of its life, to the mortality conditions observed during a given year [35]. The estimation of age-specific marriage rates used in the present analysis is situated within this framework.

The data used in the present analysis are from the latest publicly available DHS and MICS surveys for 98 countries at the time of analysis and were collected between 2006 and 2020. DHS and MICS surveys are retrospective studies of population samples. All DHS and MICS microdata files are fully anonymized before we assessed them. The surveys are based on all-women samples, including both ever-married women and never-married women. Surveys that are based on ever-married sample in which only women who were ever-married or in union were interviewed are excluded from the analysis. The present analysis focuses on women aged 15–19 years at the time of the survey; weighted sample sizes of girls aged 15–19 years ranged from 194 in Barbados to 121,533 in India.

## Estimation of age-specific marriage rates

Survival analysis, specifically, life table analysis, is applied to the latest available data on current marital status and age at first marriage among women aged 15–19 years in 98 countries to generate retrospective age-specific proportions of women who entered into marriage or consensual unions for the first time in the ages 0–17 years. The rationale for including women aged 15–19 years is that data among women 15–19 years old capture the most current incidence of child marriage at the time of the survey, while also assuring an acceptable sample size for each survey. As with the SDG summary measure, selection bias is likely to distort the results of survival analysis. However, the conditionality is mitigated to some extent because the analysis is based on the age group 15–19.

The proportions of women marrying at a single age are then converted to age-specific probabilities and conditional age-specific marriage rates for the same age range. Marriage is the event of interest. Survival time is assumed to begin at birth and ends when the individual gets married or in union before age 18. It is censored for women aged 15–17 who were still unmarried or not in union at the time of the survey.

Notationally, if x represents age in single years, the conditional rate of marriage for women at aged x—$r_x$—is estimated as follows:

$$r_x = d_x/L_x$$

where $d_X$ = number of women who entered into marriage or union *for the first time* at age $x$;

$L_x$ = number of women exposed to the risk of marriage/union between ages x and x+1. This is approximated as $.5^*(l_x + l_{x+1})$.

$l_X$ = total number of women who reach age x still unmarried. The value of $l_x$ at age 0 (the radix) can be any number. The values for ages 1 to 17 are estimated as $l_x = l_{x-1} {}^* p_{x-1}$, where $p_{x-1}$ is the probability of surviving child marriage at age x-1, i.e. remaining unmarried.

## Estimation of numbers of girls who were married in 2020

To estimate the absolute numbers of girls that were married in 2020, we assume that the age-specific probabilities and rates of child marriage that were observed in a given country at the time of the latest survey remain constant until year 2020. This assumption is based on the fact that for the majority of the 98 countries included in the study, the data used were collected between 2015 and 2020. Child marriages reported by girls aged 15–19 in a survey could have

**Table 2. Summary of differences between the direct approach and survival analysis approach.**

| Method | Computation | |
|---|---|---|
| | **Rate measure** | **Universe** |
| Direct approach | Proportion of women aged 20–24 marrying before age 18/15 | Applied to number of women aged 18/15 years |
| Survival analysis approach | Retrospective age-specific marriage rate among women aged 15–19 | Applied to number of unmarried women aged 0-17/0-14, (single year ages) |

occurred as recently as the survey year or as far back as 19 years ago, contributing to marriage rates in years prior to the study. Therefore, for countries where the data are less recent, especially prior to year 2010, for example, Azerbaijan (2006) and Bolivia (2008), the assumption made may not hold if the countries experienced significant changes in child marriage levels and marriage patterns by age in recent years. Given that child marriage rates have been declining at varying speeds in most countries in recent decades [36, 37], the assumption is likely to overestimate the number of married children in 2020 in these countries. How quickly child marriage rates have fallen over time will influence the magnitude of this overestimation.

The retrospective age-specific proportions of married women are applied to the 2019 World Population Prospects' (WPP) [38] estimates of the female population at single year ages (from 0 to age 17) in each country to derive expected numbers of women who were unmarried (at the risk of child marriage) in these countries in 2020. The age-specific marriage rates are applied to the estimated numbers of unmarried women to derive the number of women married at a given age, on the assumption that the age-specific marriage rates observed in the last DHS/MICS data apply in 2020.

The new estimates using the survival analysis are also compared with the estimates using a widely adopted direct method, referred to as "direct method" in this analysis. This direct method applies the SDG indicator "proportion of women aged 20–24 marrying before age 18" directly to the appropriate population group, in this case, the estimated number of women age 18 years in year 2020 from the 2019 WPP. A comparison of the direct method and survival analysis is presented in Table 2.

## Results

### Age-specific marriage rates

Observed marriage survival functions based on DHS and MICS data are presented in Fig 1. The data show that age-specific marriage rates vary greatly across countries. In all countries, the vast majority of girls remain unmarried until age 10. Child marriage is less prevalent in SDG region Northern America and Europe, and the highest levels in child marriage are observed in sub-Saharan African countries. In general, the rate of child marriage increases gradually until age 14 and accelerates significantly thereafter at ages 15, 16 and 17. Regional variations indicate that the rate of child marriage does not tend to escalate until 16 and beyond in European countries, relative to some sub-Saharan countries, where the rate accelerates as early as 12 years of age.

A detailed discussion of the pattern of age-specific rates of child marriage is provided below using data from 2 select countries with the highest prevalence of child marriage (Niger) and the largest number of child brides due to the overall population size of young girls (India).

Fig 2 provides country specific marriage survival curves for Niger and India. Niger has the highest child marriage rates in the world. Data obtained from the country's 2012 DHS show that for every 100 girls that reach 14 years unmarried in a given year, 17 get married before

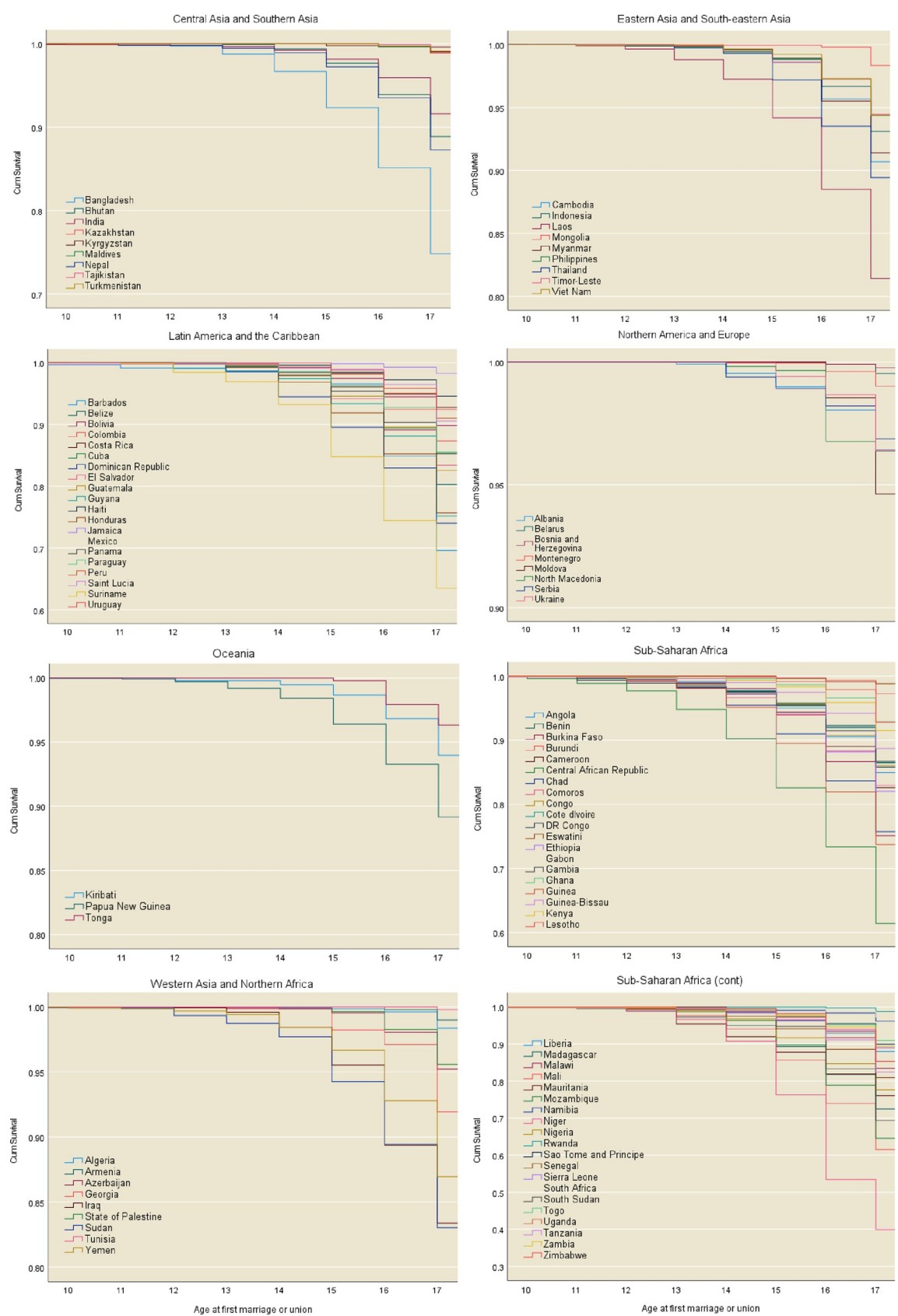

**Fig 1. Observed marriage survival functions based on the latest publicly accessible data, by SDG region.** Analysis based on weighted data (n ranged from 194 in Barbados to 121,533 in India).

their 15th birthday and this rate increases to 40 for every 100 girls between the 17th and 18th birthdays. Only 26% girls remain unmarried by age 18. This is in stark contrast to the situation in India, which has the largest absolute number of child brides, due to overall population size and age structure. Compared to Niger, age-specific marriage rates in India are much lower. Data from the India National Family Health Survey (NFHS) 2015–16 show that 84% girls reach age 18 without being married and the age-specific marriage rate ranges from 1 child bride for every 100 girls between the 14th and the 15th birthdays to 9 brides for every 100 girls between the 17th and 18th birthday. Age-specific marriage rates also indicate that marriage happens at relatively older ages in India with a peak in the number of child brides around age 17, compared to a median survival time of 16.26 years in Niger.

## Estimation of numbers of child brides

For the 98 LMICs covered in this study, survival analysis is also used to estimate the absolute number of girls marrying before age 18 in 2020. Differences between the new estimates using the survival analysis and the estimates using the direct method are assessed.

In the 98 LMICs, the survival method yields an estimated number of girls marrying before age 18, of 7,076 thousand, compared to 9,779 thousand using the direct method. A detailed comparison of the estimates is conducted to better understand the difference between estimates at the national level. The estimated numbers of girls marrying before age 18 and before age 15 using both survival analysis and the direct approach at the national level are shown in S1 Table. A sample of 17 countries are selected for the detailed comparison of estimated numbers of girls marrying before age 18 based on the following criteria: 1) the difference between

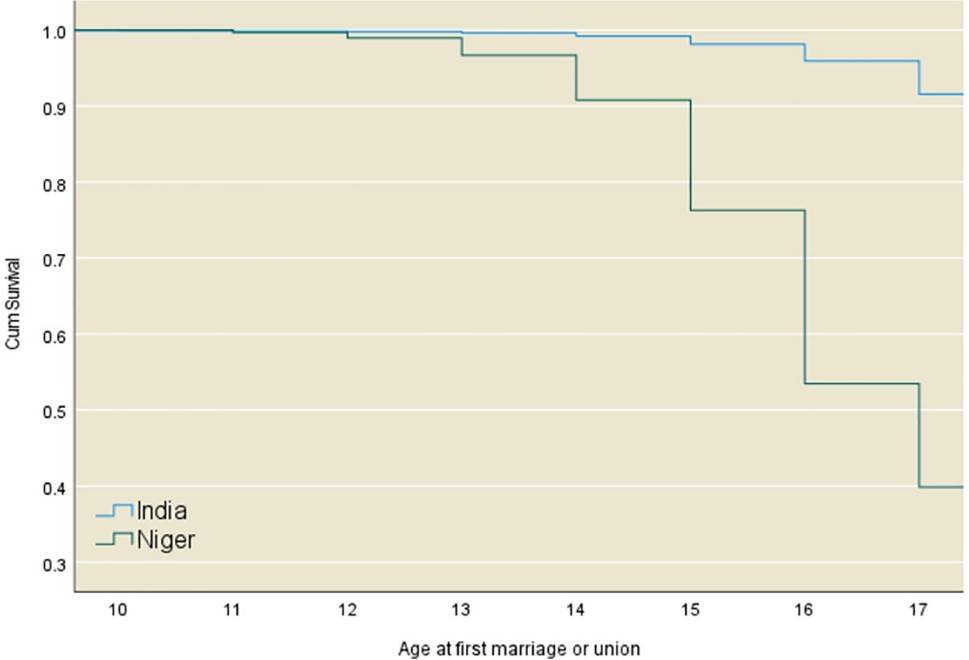

**Fig 2. Observed marriage survival functions in Niger and India, latest data.** Analysis based on weighted data (n ranged from 1830 in Niger to 121,533 in India).

the estimates is more than 10,000 if the direct estimate is higher, or more than 1,000 is the survival analysis estimate is higher; and 2) the survival analysis approach results in an estimate that is 30% lower than the estimate using the direct approach, or at least 1% higher. The 17 countries selected include Burundi, Ghana, Chad, Nepal, India, Philippines, Yemen, Ethiopia, Kenya, Zambia, Thailand, Nigeria, Bangladesh, and Indonesia which have higher direct approach estimates, and Tanzania, Mozambique, and South Africa, with higher survival analysis approach estimates (see Table 3).

As indicated in the methodology section above, the survival analysis approach is based on retrospective age-specific proportions of women who were married or entered into consensual unions for the first time in the ages 0–17 years using data collected on women aged 15–19, while the direct approach uses the marriage data collected on women aged 20–24 (see Table 2). As a result, varying child marriage rates across cohorts resulting from the rate of change in child marriage over time is likely to affect the difference in estimates resulting from the two approaches. We calculate the proportions of women marrying before age 18, among age groups 18–19, 20–24, and 25–29 years for comparison purpose (Fig 3). In the 14 countries with higher estimates using the direct approach (lower survival estimates), proportions of women aged 18–19 years marrying before age 18 are significantly lower than the proportions of women aged 20–24 years marrying before age 18, ranging from 14% lower in Bangladesh to 51% lower in Burundi, indicating possible rapid declines in the incidence of child marriage across cohorts. In comparison, in the 3 countries with higher survival estimates, we found that the child marriage proportions are unchanged or slightly higher, in age group 18–19 than age group 20–24, ranging from 5% higher in Tanzania to 42% higher in South Africa.

The difference in survival analysis estimates and estimates using the direct approach at the country level are also driven by the different ways that these two approaches accommodate changes in age structure and population trends in the country. Average annual rates (AAR) of population change between 2015 and 2020 are presented in Table 3, indicating that the AAR

**Table 3. Comparison of estimates of girls marrying before age 18 in 2020 using direct approach and survival analysis approach, in 17 select countries.**

| Country | Married before 18, among women age 20–24 (%) | Direct approach | Survival analysis approach | Difference between results (000) | Percentage difference in results (%) | Average annual rate of population change (%) |
|---------|---|---|---|---|---|---|
|  |  | Estimated number of girls marrying before age 18 (000) |  |  |  | 2015–2020 |
| Burundi | 19 | 22 | 8 | -14 | -62 | 3.15 |
| Ghana | 19 | 58 | 34 | -24 | -41 | 2.19 |
| Chad | 61 | 108 | 65 | -44 | -40 | 3.04 |
| Nepal | 33 | 106 | 64 | -41 | -39 | 1.51 |
| India | 25 | 2987 | 1905 | -1082 | -36 | 1.04 |
| Philippines | 17 | 166 | 107 | -58 | -35 | 1.41 |
| Yemen | 32 | 98 | 64 | -35 | -35 | 2.37 |
| Ethiopia | 40 | 511 | 332 | -179 | -35 | 2.62 |
| Kenya | 23 | 133 | 87 | -47 | -35 | 2.32 |
| Zambia | 29 | 60 | 39 | -21 | -34 | 2.93 |
| Thailand | 20 | 88 | 60 | -27 | -31 | 0.31 |
| Nigeria | 43 | 902 | 631 | -271 | -30 | 2.59 |
| Bangladesh | 51 | 785 | 551 | -234 | -30 | 1.05 |
| Indonesia | 16 | 371 | 261 | -110 | -30 | 1.14 |
| Tanzania | 31 | 187 | 189 | 2 | 1 | 2.97 |
| Mozambique | 48 | 167 | 173 | 6 | 3 | 2.90 |
| South Africa | 4 | 17 | 18 | 1 | 8 | 1.37 |

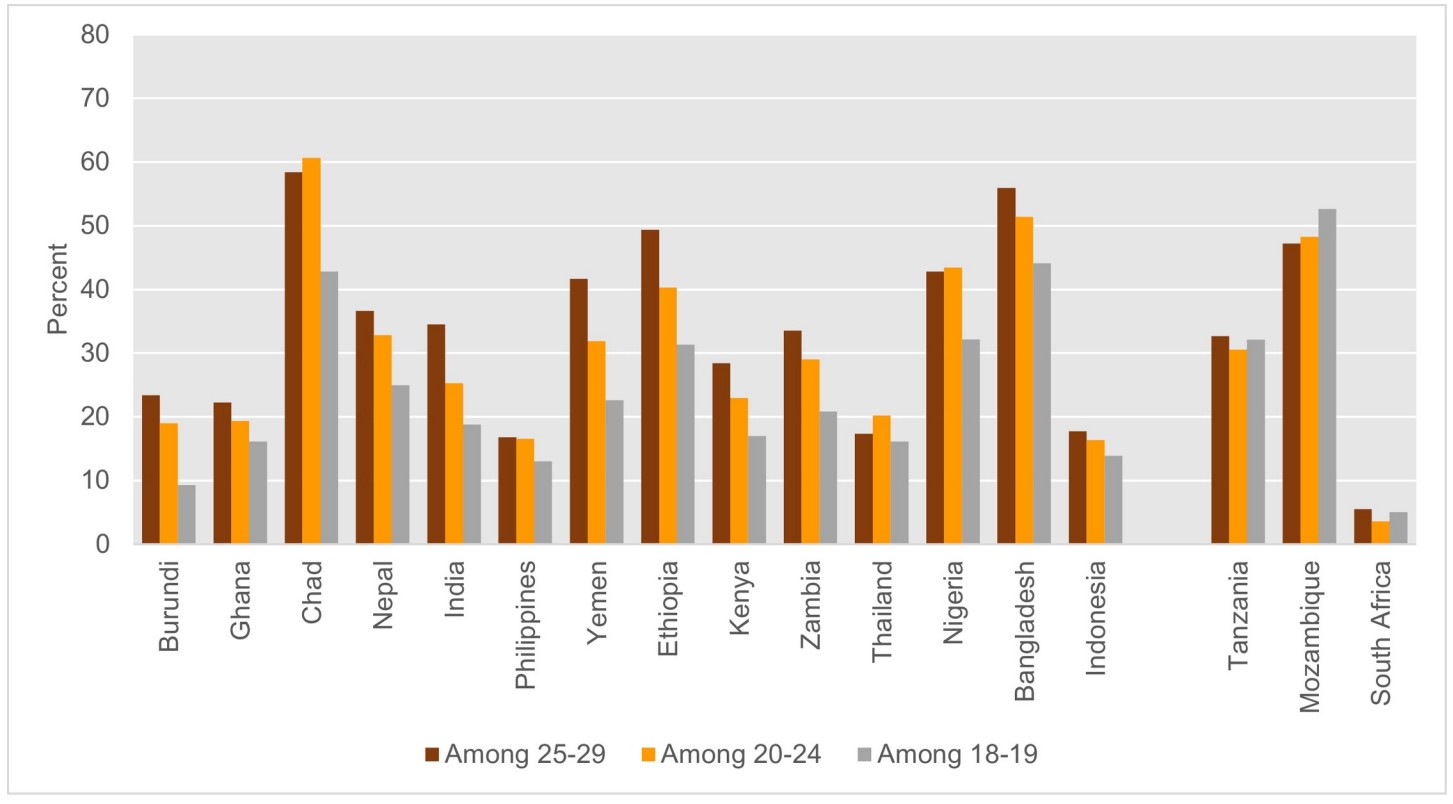

**Fig 3. Proportion of women marrying before age 18, among women in age groups 18–19, age 20–24, and age 25–29 years, in select 17 countries.**

between 2015 and 2020 ranges from 3.15% in Burundi to 1.04% in India and 0.31% in Thailand.

To better understand the contribution of age structure to our estimates in the 17 countries, Fig 4 presents the number of women aged 0–24 years by single-year age in each country. The 17 countries have vastly different age structures. In India, female population at age 0 is 11,505 thousand compared to 11,871 thousand for female population at age 17. In comparison, female population at age 0 in Mozambique is 540 thousand, 1.5 times the size of the population at age group 17, which is 358 thousand.

Because the survival analysis approach uses both single-year-age-specific child marriage hazard rates among girls aged 15–19 and the age structure of the population, it better captures how recent changes in child marriage rates and population growth in a country is impacting the number of girls at risk of early marriage. Driving by rapid decline in the incidence of child marriage, the survival analysis may result in lower estimates of child marriage in both countries that have lower rates of population growth and transitional age structures resulting from near constant but low fertility rates (in India, for example), and countries with high rates of population growth (in Burundi and Chad, for example). In countries with constant or slightly rising child marriage rates, the survival analysis yields higher estimates of child marriage in both countries with high rates (in Mozambique, for example) and low rates of population growth (in South Africa, for example). When using single-year-age-specific hazard rates among girls aged 18–19 for the survival analysis approach and the proportions of 18-19-year-olds marrying before age 18 for the direct approach, the survival analysis almost always produces higher estimates in countries with high rates of population growth due to sustained high

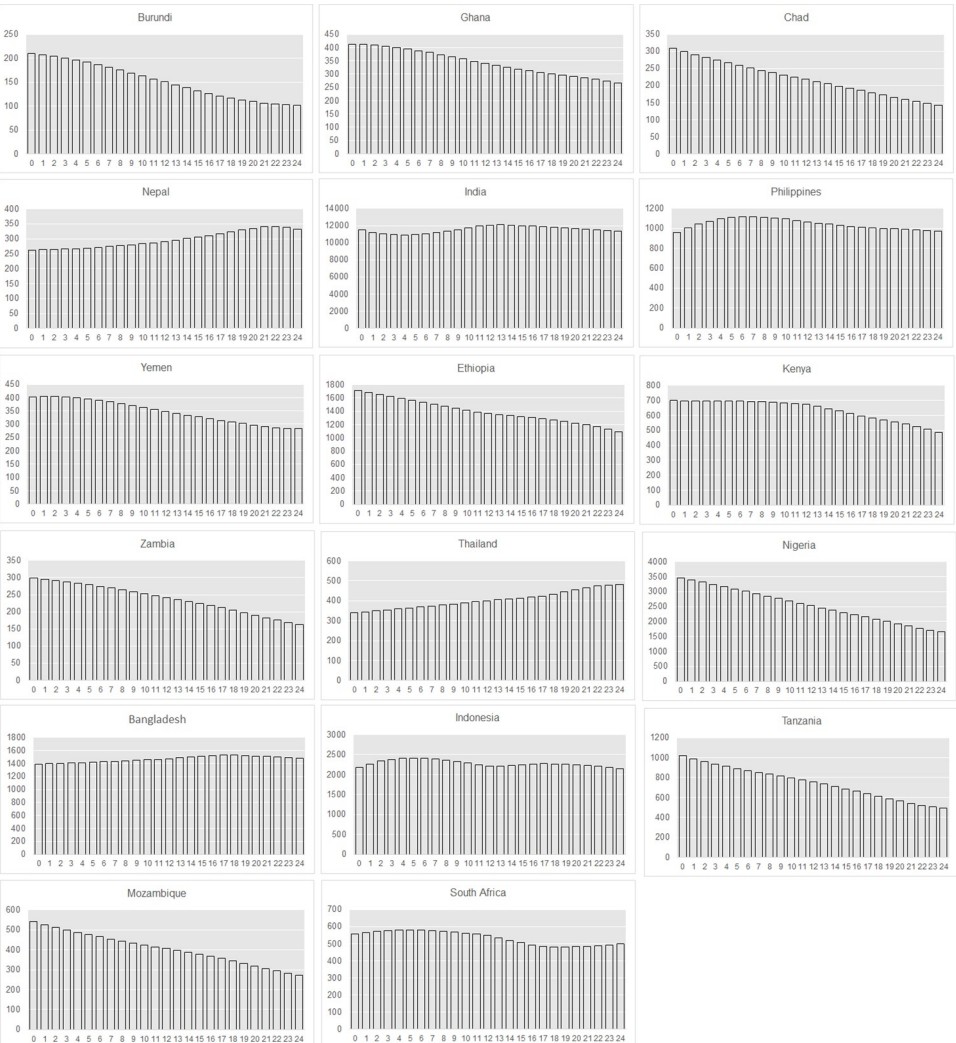

**Fig 4. Age structure–number of women age 0–24 years (thousands), by single-year age, in 17 select countries, year 2020.**

fertility rates, very young (broad-based) age structures, and relatively high child marriage rates; and similar or lower estimates in countries with lower rates of population growth and transitional age structures resulting from low fertility rates as compared to the direct approach (data not shown).

## Discussion

The overarching objective of this study was to present an alternate analytical approach for measuring the levels in child marriage and estimating the number of girls that are affected by child marriage. Currently, the widely used approach involves calculating the proportion of women aged 20–24 years who were married or entered into an informal union before age 15 and before age 18, which is also the official SDG indicator measuring the issue of child marriage (SDG 5.3.1). However, this indicator is only a summary statistic of the prevalence of child marriage, it does not provide a full picture of the dynamics of the incidence of marriage among children at different ages before they reach age 18. In this paper, we showed that by

applying survival analysis techniques we can gain a better understanding, not only of the age patterns in child marriage rates, but potentially more precise estimates of the absolute numbers of girls at risk of child marriage.

The analytical power provided by survival analysis techniques in the study of child marriage is conceptually appealing from a policy and program point of view because it equips policy makers and program managers with granular information about age-specific risks in individual countries, and the changing numbers of girls at risk where population growth is especially high. Survival analysis has been previously applied in various life events [39–41]. A similar approach has recently been applied to highlight age-specific risks of female genital mutilation (FGM), and the extent to which the pace of population growth in affected countries is changing the absolute numbers of girls at risk [42]. In both analyses, the rising absolute numbers of girls at risk is of crucial importance to policymakers, highlighting where accelerated interventions are needed to eliminate these practices by 2030.

A recent analysis of trends in 31 sub-Saharan African countries found that child marriage has decreased over time, but the decline has been concentrated among older adolescents [36]. This alternative approach allows for better understanding of trends in child marriage over time and is sensitive enough to detect recent declines, both small and large, to provide information for impact measurement of program interventions. More precise estimates of the number of girls affected also allow more efficient resource allocation.

More importantly, analyzing marriage by single-year ages allows us to see when child marriage accelerates, and thus to design and implement target interventions before the acceleration. From the results presented, it is clear that intervention around age 10, right before puberty, is critical in many high-prevalence countries. Relevant interventions include supporting girls to complete primary schooling and transition to lower secondary school, giving girls critical life skills including self-confidence and relationship skills, disseminating knowledge of their health as well as their rights, providing financial literacy, etc. [43, 44]. For example, the Empowerment and Livelihood for Adolescents (ELA) program in Uganda, which aimed to empower adolescent girls by providing 1) vocational training to start small-scale income-generating activities, and 2) life skills training and information on sex, reproduction, and marriage showed significant improvements in girls' entrepreneurial skills, increased girls' knowledge of and control over their bodies in terms of childbearing, relationship quality, and sexual intercourse, and changed girls' aspirations of marriage and childbearing. Notably, girls in intervention communities were 6.9 percentage points less likely to be married or cohabiting two years later [45]. Where marriage accelerates at age 15, supporting girls to stay in school and complete secondary schooling is critical, in addition to supporting their access to sexual and reproductive health services, and providing them with financial services such as access to savings accounts. Young married girls who may be at particularly high risk of poor sexual and reproductive health outcomes will also require more targeted attention, separately from unmarried girls.

Evidence show that consistent minimum marriage age laws offer some protection against the exploitation of girls [46]. Data on age patterns of child marriage can serve as basis for assessing the implementation of existing minimum marriage age laws and advocating for such laws in countries where it is absent.

Although the study reached its aim of estimating age-specific marriage rates among girls before age 18 and the absolute numbers of girls that were affected by child marriage in 2020 in the 98 countries for which data were accessible, a few limitations need to be highlighted. First, data for the vast majority of countries included in the analysis were collected before the reference year 2020. Thus, if these countries experienced significant changes in the age pattern of child marriage in recent times, the assumptions made when estimating absolute numbers of

girls affected by child marriage for 2020 may not hold. In light of the fact that child marriage has been decreasing at different rates in most countries over the last few decades [36, 37], the assumption is likely to overstate the estimated number of married girls in these countries in 2020. However, the estimated number of girls marrying before reaching the age of 18 using the direct approach is based on the same assumption and is likely to be inflated. Because of this, the comparison between the two approaches is considered to be appropriate.

Second, the results of the study are sensitive to the common measurement issues that are discussed in this paper. Errors in reporting on age at first marriage may include recall bias, social desirability bias, and survey completion issues. Recall bias occurs when individuals are unable to recall prior events or their timing. It can be more prevalent in older cohorts than in younger cohorts, since more time has passed between the survey and the occurrence of child marriage [47]. People's understanding of "marriage" and "being married" may change over time due to social forces advocating for women's empowerment, as well as anti–child marriage campaigns and the introduction or reform of minimum-age-at-marriage legislation [25, 36]. Thus, younger cohorts, such as females aged 15–19 included in the survival analysis, may feel more pressure to claim to be older at the time of their marriage than they really were, in comparison to older cohorts, such as females aged 20–24 included in the direct approach. However, these biases should be contextualized; for instance, understatement in age at first marriage among women in Bangladesh was identified perhaps due to the societal acceptability of early marriage for women in that country [27]. Assessment of the quality of age reporting in DHS surveys shows that women's reporting on age at first marriage are more incomplete than other age reporting primarily due to missing month [24]. DHS performs extensive data imputations to address incomplete and inconsistent reporting [48]. While imputation on age at first marriage could lead to biased results, evidence show that when an age at first marriage is incomplete, it is typically because the age and year of birth were given but not the month, or the month was discordant with the year and age, necessitating the month's imputation or modification. It is uncommon for age to need imputation [24].

Despite these limitations, this analysis makes an important contribution to the measurement of child marriage. In order to address the limitations associated with the collection and assessment of timely data on child marriage, we have the following recommendations. Countries should aim to strengthen their population data ecosystems to have fully functional CRVS systems that are complemented by regular census and household survey undertakings, as part of efforts to track progress in achieving the SDGs. Although the two statistical systems produce two different perspectives in respect to marriage–with censuses and household surveys presenting the household admission of who is married, and the administrative data presenting the state's record of who is married in terms of the prevailing laws of the country–a national statistics system that is made up of regular censuses and household surveys, plus a fully functional CRVS system covering all marriage types is desirable.

Censuses should adopt the more comprehensive household survey approach of measuring marital status. Specifically, informal union should be recognized, and data on age at first informal union should be systematically collected. Furthermore, standard definitions of marriage and union should be adopted across censuses and surveys. In the 2020 census round currently underway, questions on the age at first marriage, and on the registration status of marriages by type are recommended [49, 50]; similar recommendations are applicable to relevant household surveys. Additionally, efforts should be made to improve data collection tools in order to minimize misreporting, as well as to develop techniques for correcting estimates based on self-reported data of women sampled in household surveys.

In conclusion, the study demonstrates an alternative method of estimating levels of child marriage using survival analysis approach and suggests that accounting for both single-year-

age-specific child marriage hazard rates and the population age structure results in lower figures in countries with rapid decrease in child marriage and higher figures in countries with constant or slightly rising child marriage rates relative to the direct approach. The study highlights the program implications of precise single-year-age-specific child marriage hazard rates and advises that child marriage programs should target girls as young as age 10 in high-prevalence countries. Future research using the survival analysis approach to assess the evolution of child marriage should be prioritized. We recommend that the survival analysis methodology be more widely adopted for various progress monitoring and programme design purposes.

## Supporting information

**S1 Table. Proportion of women age 20–24 marrying before age 15 and before age 18, and estimated number of girls marrying before age 15 and before age 18 using survival analysis approach and direct approach, latest data in 98 countries.**
(DOCX)

## Acknowledgments

The authors would like to thank Goleen Samari for her careful reading of the manuscript and her insightful comments and suggestions.

## Author Contributions

**Conceptualization:** Mengjia Liang.

**Data curation:** Mengjia Liang.

**Formal analysis:** Mengjia Liang, Sandile Simelane.

**Methodology:** Mengjia Liang, Sandile Simelane.

**Supervision:** Rachel Snow.

**Visualization:** Mengjia Liang.

**Writing – original draft:** Mengjia Liang, Sandile Simelane, Satvika Chalasani.

**Writing – review & editing:** Satvika Chalasani, Rachel Snow.

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
