## [Decision Letter · Decision Letter 0]

13 Jul 2021

PONE-D-21-16939

New Estimations of Child Marriage: Evidence from 91 Low- and Middle-Income Countries

PLOS ONE

Dear Dr. Liang,

Thank you for submitting your manuscript to PLOS ONE. After careful consideration, we feel that it has merit but does not fully meet PLOS ONE’s publication criteria as it currently stands. Therefore, we invite you to submit a revised version of the manuscript that addresses the points raised during the review process.

We look forward to receiving your revised manuscript.

Kind regards,

Bidhubhusan Mahapatra, Ph.D.

Academic Editor

PLOS ONE

Journal Requirements:

Additional Editor Comments:

This is an important body of work. As highlighted by both the reviewers, there are some methodological concerns that authors must address.

Reviewers' comments:

Reviewer's Responses to Questions

**Comments to the Author**

1. Is the manuscript technically sound, and do the data support the conclusions?

Reviewer #1: Partly

Reviewer #2: Yes

2. Has the statistical analysis been performed appropriately and rigorously? 

Reviewer #1: Yes

Reviewer #2: Yes

3. Have the authors made all data underlying the findings in their manuscript fully available?

Reviewer #1: Yes

Reviewer #2: Yes

4. Is the manuscript presented in an intelligible fashion and written in standard English?

Reviewer #1: Yes

Reviewer #2: Yes

5. Review Comments to the Author

Reviewer #1: The authors used descriptive survival analyses to estimate age-specific risks for marriage in 91 countries based on DHS and MICS data. Their approach is an alternative way to quantify child marriage practices that has some advantages and disadvantages relative to other commonly used metrics. At this stage, I find that the authors have not sufficiently described their methods and have not thoughtfully considered sources of bias and their impact on their estimates. The manuscript needs to be strengthened in these regards before it is suitable for publication and could use editing for clarity throughout. My comments on specific sections of the paper follow.

Measurement of child marriage

The authors cite a source from 1993 to indicate a common refrain: that marriage in sub-Saharan Africa “is a process”. This assertion is problematic for many reasons. First, it implies that marriage practices have remained static for at least three decades. Second, it implies that marriage practices across this large and diverse region are uniform. Third, it implies that marriage processes in the region differ from processes elsewhere, which is somewhat misleading. For example, in North America it is common to become engaged and/or have children with a partner far in advance of a legal or religious ceremony. In China, legal registration often occurs substantially later than a ceremony. (Clearly, this is bit of a thorn in this reviewer’s side.) I recommend that the authors omit this outdated statement and simply acknowledge that perceptions of marital status vary across the societies. This would also seem more in line with the scope of this paper, which includes 91 countries from across the globe; the focus on Africa here seems unnecessary.

Incompleteness of reporting on age at marriage is an important data quality issue (lines 108-118) but the authors omit another equally important concern: the accuracy of the ages reported. The approach used in this paper assumes that, among those who did report how old they were at the time of their first marriage/cohabitation, the ages provided are perfect and unaffected by recall bias or social desirability bias. This is unlikely and its potential impact on the estimates in this study should be discussed. In addition, the DHS often imputes ages that are missing. What affect might that have on the measurement approach used in this study?

The authors include the language of the questions included in the DHS after line 160. The data used in their analyses cover the most recent three phases of the DHS women’s questionnaire (Phases 5-7). Has the language of these questions remained consistent across these phases? Is the language used in MICS surveys exactly the same?

Methods

In the first step of their analyses, the authors estimated the age-specific risk of child marriage in each country based on data from the most recent DHS or MICS. The ideal estimate would have been a conditional probability based on longitudinal data, i.e., conditional on remaining alive and unmarried until the day before your 12th birthday, what is the probability that you get married or begin cohabiting between your 12th birthday and the day before your 13th birthday? However, the structure of the data they are relying upon creates a few problems with this approach. First, the sample includes only those aged 18-19. This means that the probabilities they estimate are also conditional on having survived until the age of 18-19. This creates a selection bias. If girls who marry at very early ages are less likely to survive the experience, possibly as a result of the harms noted in the introduction, and die before reaching the age of 18, they will be omitted from these estimates. (This bias affects summary measures as well, which are conditional on surviving until the ages of 20-24, but should be discussed nonetheless.) The Kaplan Meier methodology that the authors use can handle censoring, so why not include all girls between 15 and 18 in the sample? This would reduce this conditionality to some degree.

Second, the time frame for the age-specific marriage rates is somewhat tricky to pin down. The authors suggest that these are the rates in the year of the most recent survey, but that isn’t really true. For example, an 18-year-old respondent to a 2010 DHS who reported that she married at the age of 13 is actually contributing to a marriage rate in 2005, since her marriage actually occurred five years prior. Again, this needs to be discussed and its implications for overestimation (see next paragraph) made transparent.

In the second step of their analyses, the authors applied the age-specific conditional probability estimates obtained for each country to estimates of the country’s population in 2017 to estimate the number of girls married in that year. They aptly note that this approach assumes that the age-specific probabilities remained unchanged between the year the marriages occurred and 2017. However, I would like to see more justification for the claim that this “…is a fairly reasonable assumption for the majority of the 91 countries included in the study because the data used were collected between 2010 and 2016.” They then raise concerns about countries in which the most recent survey was conducted prior to 2010. Is there something important about the distinction between 2009/2010? Likely not. I would also like to see the authors be more transparent about how this assumption is likely to affect their estimates. With very few exceptions, child marriage rates have been falling at different speeds across the globe in recent decades. Basing these counts on marriage rates from earlier years will therefore result in overestimates of the number of married children in 2017. The magnitude of this overestimation will vary depending on how quickly the nation’s child marriages rates have been falling over time.

I am confused by the language in lines 223-229, which seems to indicate that the age-specific probabilities derived from the first stage of the analysis were applied to the population estimates from the WPP twice. There would seem to be no need to use a two-stage estimation process at this point. For example, applying the probability of marriage at the age of 13 in Nigeria to the total number of 13-year-olds living in Nigeria in 2017 would give you an estimate of the number of married 13-year-olds in that year. If the authors did this differently, by first estimating the number of unmarried girls and then applying the age-specific probabilities to the estimated number of unmarried girls, this would seem to double count. This must be clarified.

Results

Figures 1 and 2 need work. First of all, probabilities cannot take values less than zero, but the x-axis includes -1. This needs to be corrected. Most of the space in this figure is wasted since marriage prior to the age of 10 is very rare. (And when it occurs in the data, is quite likely the result of inaccurate reporting or data entry error.) One way to adjust this would be to begin the risk period at the age of 10. Why are there vertical lines at 10 and 14 years? The right scale includes residual code that should be cleaned up (“Country code for naeu”, etc.). There are no countries from North America included in this study; the figure label and the text of the manuscript should be corrected to accurately reflect that.

Use of the term “hazard rates” (line 259, others) is somewhat confusing. This term has a precise definition associated with Cox proportional hazard models for survival analysis, which the authors did not use. I recommend the authors stick with the language of rates or conditional probabilities.

On Lines 329-354, the authors suggest that observed differences in the estimated number of girls married between the “survival analysis” and “direct” measurement approaches are attributable to changes in the age structure of populations within countries over time. As far as I can tell from what is written in the methods section of the paper, the authors estimated the number of girls married by simply multiplying the number of girls at each individual age by the age-specific marriage risk. To illustrate, if the risk of marriage at the age of 13 in India based on the 2015-16 NFHS was 0.03% and there were 1 million 13-year-old girls living in India in 2017 according to the WPP, they would estimate a total of 300 married 13-year-olds in 2017 (1 million * 0.0003 = 300). This process would be repeated for 14-year-olds, 15-year-olds, etc. and summed to estimate a total number of married girls. As described above, this would almost certainly lead to overestimates of the number of girls married simply because age-specific marriage rates from earlier years are being applied to population estimates in later years, and in most contexts child marriage is becoming less common over time. Of course, if the population is growing, it is possible that the absolute number of married girls could increase over time even if the rates of child marriage are falling. However, it would be very challenging to parse apart how much of the observed differences in these numbers is attributable to the bias and how much is attributable to population growth. This is not discussed at all. I also wonder if the countries in which differences between estimation approaches were minimal were those in which the survey was conducted closer to 2017. (For example, the Indian NFHS was conducted in 2015-16 which would make estimates based on it likely more accurate for application to population estimates from 2017.) I would like to see the authors investigate this possibility.

There are no quantitative measures of the uncertainty around these estimates, such as 95% confidence intervals, anywhere in the paper. This would help with interpretation of differences between estimation strategies, such as in Figure 3. This is especially odd given that the authors note that limiting their sample to 18-19-year-olds reduced statistical power and widens confidence intervals (lines 419-421).

Given the biases that affect the estimates presented in this study and the fact that no validation study was conducted (and indeed would be very challenging to conduct), claiming that these estimates are “more precise” (lines 366, 383), “more reliable” (line 448) or “more accurate” (line 33) than other approaches to quantifying child marriage is an unjustified overstatement. All quantitative measures have pros and cons. Calculation of age-specific marriage rates is useful and the Kaplan Meier curves present more detailed information on the timing of marriage than is communicated in summary statistics. This is an important contribution and may be informative for context-specific interventions in this domain. However, the authors have neglected some important limitations of their approach relative to others. For example, they require strong assumptions about the stability of child marriage rates over time and are less readily interpretable by broad audiences. Greater transparency and humility about our imperfect attempts to measure this phenomenon would make this a much stronger paper.

Reviewer #2: This paper makes a major contribution to the field by proposing an alternative measurement of child marriage that captures recent estimates as against the standard indicator of proportion of 20-24 year-old women married by age 15/18 that has been often seen as capturing the 'incidence' of child marriage in the past. I would like to flag a few points to the attention of the authors.

1. Authors note that estimates of child marriage are produced for the year 2017, providing baseline estimates of child marriage at the dawn of the SDG tracking period which is a good rationale. However, I am wondering whether the authors could also provide an estimate for the year 2020 as it can help monitor the progress towards eliminating child marriage in the five years since the declaration of the SDGs. In fact, 23 of the 91 countries included in the study have survey data that cover the period between 2016-2020 at least from the DHS.

2. Authors have included 91 LMICs and a scan of DHS website suggests some important omissions, for example, South Africa, Tanzania, Guatemala and there are others too (Appendix Table 1), despite having data from DHS for the period that authors have relied for the analysis. I am wondering whether authors applied any exclusion criteria.

3. Authors note that to estimate the absolute numbers of girls that were married in 2017, they have assumed that the age-specific probabilities and rates of child marriage that were observed in a given country at the time

of the latest survey remain constant until year 2017, and that this was a fairly reasonable assumption for the

majority of the 91 countries included in the study. Trend data from DHS suggest that some of the included countries may have experienced noticeable decline (and in rare cases, an increase) even during a short-span of 5-6 years. I am wondering whether the authors could have projected different scenario based different level of rate of child marriage.

4. Authors have discussed the conceptual, methodological and data quality issues related to nuptiality data and authors have acknowledged this as a limitation. It would have been good if authors could reflect more on how such issues, particularly age misreporting may affect their estimated number of girls marrying before age 18;

5. Authors argue that data on age patterns of child marriage can serve as basis for assessing the implementation of existing minimum marriage age laws and advocating for such laws in countries where it is absent. While such laws can serve as a deterrent, marriage age misreporting around the legal age cannot be ignored.

6. Could authors also provide the estimated number of girls marrying before age 15 as an annexure?

7. Could authors include regional estimates as well?

8. Although child marriage is more prevalent among girls than boys, could the authors have estimated the incidence of child marriage among boys too?

6. PLOS authors have the option to publish the peer review history of their article (what does this mean?). If published, this will include your full peer review and any attached files.

Reviewer #1: **Yes: **Alissa Koski

Reviewer #2: No

---

## [Author Response · Author response to Decision Letter 0]

27 Aug 2021

Dear Dr. Mahapatra, Dr. Koski and peer reviewer:

Thank you for the opportunity to submit a revised draft of our manuscript titled New Estimations of Child Marriage: Evidence from 98 Low- and Middle-Income Countries to PLOS ONE. We are grateful to the reviewers for their insightful comments on our paper. We have been able to incorporate changes to reflect most of the suggestions provided by the reviewers.

Please see below is a point-by-point response to the reviewers’ comments and concerns.

Comments from Reviewer 1

Reviewer #1: The authors used descriptive survival analyses to estimate age-specific risks for marriage in 91 countries based on DHS and MICS data. Their approach is an alternative way to quantify child marriage practices that has some advantages and disadvantages relative to other commonly used metrics. At this stage, I find that the authors have not sufficiently described their methods and have not thoughtfully considered sources of bias and their impact on their estimates. The manuscript needs to be strengthened in these regards before it is suitable for publication and could use editing for clarity throughout. My comments on specific sections of the paper follow.

Measurement of child marriage

The authors cite a source from 1993 to indicate a common refrain: that marriage in sub-Saharan Africa “is a process”. This assertion is problematic for many reasons. First, it implies that marriage practices have remained static for at least three decades. Second, it implies that marriage practices across this large and diverse region are uniform. Third, it implies that marriage processes in the region differ from processes elsewhere, which is somewhat misleading. For example, in North America it is common to become engaged and/or have children with a partner far in advance of a legal or religious ceremony. In China, legal registration often occurs substantially later than a ceremony. (Clearly, this is bit of a thorn in this reviewer’s side.) I recommend that the authors omit this outdated statement and simply acknowledge that perceptions of marital status vary across the societies. This would also seem more in line with the scope of this paper, which includes 91 countries from across the globe; the focus on Africa here seems unnecessary.

Response: Thank you for the comment. We use sub-Saharan Africa as an example to demonstrate the perceptions of marriage vary across the societies. However, in order to avoid an emphasis on the SSA, we have removed the statement as suggested.

Incompleteness of reporting on age at marriage is an important data quality issue (lines 108-118) but the authors omit another equally important concern: the accuracy of the ages reported. The approach used in this paper assumes that, among those who did report how old they were at the time of their first marriage/cohabitation, the ages provided are perfect and unaffected by recall bias or social desirability bias. This is unlikely and its potential impact on the estimates in this study should be discussed. In addition, the DHS often imputes ages that are missing. What affect might that have on the measurement approach used in this study?

Response: We appreciate the comment. We have included detailed discussions in both the background and discussion sections about the accuracy of the ages reported and the incompleteness of reporting, as well as their implications on the estimates.

The authors include the language of the questions included in the DHS after line 160. The data used in their analyses cover the most recent three phases of the DHS women’s questionnaire (Phases 5-7). Has the language of these questions remained consistent across these phases? Is the language used in MICS surveys exactly the same?

Response: Thank you. Standard questions used for child marriage measurement in both DHS and MICS women’s questionnaires are now included in Table 1. 

Methods

In the first step of their analyses, the authors estimated the age-specific risk of child marriage in each country based on data from the most recent DHS or MICS. The ideal estimate would have been a conditional probability based on longitudinal data, i.e., conditional on remaining alive and unmarried until the day before your 12th birthday, what is the probability that you get married or begin cohabiting between your 12th birthday and the day before your 13th birthday? However, the structure of the data they are relying upon creates a few problems with this approach. 

First, the sample includes only those aged 18-19. This means that the probabilities they estimate are also conditional on having survived until the age of 18-19. This creates a selection bias. If girls who marry at very early ages are less likely to survive the experience, possibly as a result of the harms noted in the introduction, and die before reaching the age of 18, they will be omitted from these estimates. (This bias affects summary measures as well, which are conditional on surviving until the ages of 20-24, but should be discussed nonetheless.) The Kaplan Meier methodology that the authors use can handle censoring, so why not include all girls between 15 and 18 in the sample? This would reduce this conditionality to some degree.

Response: Thank you for pointing this out. Selection bias is now discussed in both the background and methods sections. We chose the age group of 18-19 for the survival analysis in order to make a comparison to the second variant of the direct approach, namely the proportion of women aged 18-19 who married before reaching the age of 18. However, we agree that adding the years of 15-17 offers a number of benefits, which is made feasible by the life table analysis method we employed. Thus, we revised our survival analysis approach to include 15-17-year-old females.

Second, the time frame for the age-specific marriage rates is somewhat tricky to pin down. The authors suggest that these are the rates in the year of the most recent survey, but that isn’t really true. For example, an 18-year-old respondent to a 2010 DHS who reported that she married at the age of 13 is actually contributing to a marriage rate in 2005, since her marriage actually occurred five years prior. Again, this needs to be discussed and its implications for overestimation (see next paragraph) made transparent.

Response: Thank you. This issue is now discussed in both the methods and discussion sections 

In the second step of their analyses, the authors applied the age-specific conditional probability estimates obtained for each country to estimates of the country’s population in 2017 to estimate the number of girls married in that year. They aptly note that this approach assumes that the age-specific probabilities remained unchanged between the year the marriages occurred and 2017. However, I would like to see more justification for the claim that this “…is a fairly reasonable assumption for the majority of the 91 countries included in the study because the data used were collected between 2010 and 2016.” They then raise concerns about countries in which the most recent survey was conducted prior to 2010. Is there something important about the distinction between 2009/2010? Likely not. I would also like to see the authors be more transparent about how this assumption is likely to affect their estimates. With very few exceptions, child marriage rates have been falling at different speeds across the globe in recent decades. Basing these counts on marriage rates from earlier years will therefore result in overestimates of the number of married children in 2017. The magnitude of this overestimation will vary depending on how quickly the nation’s child marriages rates have been falling over time.

Response: Thank you for your feedback. The language regarding the assumption has been revised in accordance with the suggestions, and the issue of overestimation has now been discussed in both the methods and the discussion sections.

I am confused by the language in lines 223-229, which seems to indicate that the age-specific probabilities derived from the first stage of the analysis were applied to the population estimates from the WPP twice. There would seem to be no need to use a two-stage estimation process at this point. For example, applying the probability of marriage at the age of 13 in Nigeria to the total number of 13-year-olds living in Nigeria in 2017 would give you an estimate of the number of married 13-year-olds in that year. If the authors did this differently, by first estimating the number of unmarried girls and then applying the age-specific probabilities to the estimated number of unmarried girls, this would seem to double count. This must be clarified.

Response: Thank you for your feedback. The conditional rate of first marriage at age x is applied to number of never married women at age x because they are the population at risk of first marriage at age x. Ever married women at age x were counted by conditional rates of first marriages at previous ages. 

Results

Figures 1 and 2 need work. First of all, probabilities cannot take values less than zero, but the x-axis includes -1. This needs to be corrected. Most of the space in this figure is wasted since marriage prior to the age of 10 is very rare. (And when it occurs in the data, is quite likely the result of inaccurate reporting or data entry error.) One way to adjust this would be to begin the risk period at the age of 10. Why are there vertical lines at 10 and 14 years? The right scale includes residual code that should be cleaned up (“Country code for naeu”, etc.). There are no countries from North America included in this study; the figure label and the text of the manuscript should be corrected to accurately reflect that.

Response: Thank you. Figures 1 and 2 are re-produced to reflect the updated analysis and the suggestions. North America and Europe is cleared labeled as a SDG region. 

Use of the term “hazard rates” (line 259, others) is somewhat confusing. This term has a precise definition associated with Cox proportional hazard models for survival analysis, which the authors did not use. I recommend the authors stick with the language of rates or conditional probabilities.

Response: Thank you. We used hazard rates in the life table analysis therefore this term is used in the manuscript. However, we changed “hazard rates” to “rates” in original line 259 to make it clearer. 

On Lines 329-354, the authors suggest that observed differences in the estimated number of girls married between the “survival analysis” and “direct” measurement approaches are attributable to changes in the age structure of populations within countries over time. As far as I can tell from what is written in the methods section of the paper, the authors estimated the number of girls married by simply multiplying the number of girls at each individual age by the age-specific marriage risk. To illustrate, if the risk of marriage at the age of 13 in India based on the 2015-16 NFHS was 0.03% and there were 1 million 13-year-old girls living in India in 2017 according to the WPP, they would estimate a total of 300 married 13-year-olds in 2017 (1 million * 0.0003 = 300). This process would be repeated for 14-year-olds, 15-year-olds, etc. and summed to estimate a total number of married girls. As described above, this would almost certainly lead to overestimates of the number of girls married simply because age-specific marriage rates from earlier years are being applied to population estimates in later years, and in most contexts child marriage is becoming less common over time. Of course, if the population is growing, it is possible that the absolute number of married girls could increase over time even if the rates of child marriage are falling. However, it would be very challenging to parse apart how much of the observed differences in these numbers is attributable to the bias and how much is attributable to population growth. This is not discussed at all. I also wonder if the countries in which differences between estimation approaches were minimal were those in which the survey was conducted closer to 2017. (For example, the Indian NFHS was conducted in 2015-16 which would make estimates based on it likely more accurate for application to population estimates from 2017.) I would like to see the authors investigate this possibility.

Response: Thank you for your feedback. Both the methods and discussion sections have now addressed the issue of overestimation. Kindly note that the estimated number of females marrying before reaching the age of 18 using the direct method is also based on the same assumption and is thus likely to be overstated. As a result, the comparison of the two methods is deemed acceptable. Similarly, the year of the survey is unlikely to affect the differences between the two approaches. We have also discussed in detail how the differences in survival analysis estimates and direct estimates at the country level are caused by both varying child marriage rates across cohorts due to the rate of change in child marriage over time and the distinct ways in which these two approaches account for changes in the country's age structure and population trends.

There are no quantitative measures of the uncertainty around these estimates, such as 95% confidence intervals, anywhere in the paper. This would help with interpretation of differences between estimation strategies, such as in Figure 3. This is especially odd given that the authors note that limiting their sample to 18-19-year-olds reduced statistical power and widens confidence intervals (lines 419-421).

Response: Thank you. We expanded our sample and included 15-17-year-olds. We hope to make a comparison with data for SDG indicator 5.3.1 and they are often reported as point estimates. 

Given the biases that affect the estimates presented in this study and the fact that no validation study was conducted (and indeed would be very challenging to conduct), claiming that these estimates are “more precise” (lines 366, 383), “more reliable” (line 448) or “more accurate” (line 33) than other approaches to quantifying child marriage is an unjustified overstatement. All quantitative measures have pros and cons. Calculation of age-specific marriage rates is useful and the Kaplan Meier curves present more detailed information on the timing of marriage than is communicated in summary statistics. This is an important contribution and may be informative for context-specific interventions in this domain. However, the authors have neglected some important limitations of their approach relative to others. For example, they require strong assumptions about the stability of child marriage rates over time and are less readily interpretable by broad audiences. Greater transparency and humility about our imperfect attempts to measure this phenomenon would make this a much stronger paper.

Response: Thank you for pointing this out. We have adjusted the languages as per the suggestions. 

Reviewer #2: This paper makes a major contribution to the field by proposing an alternative measurement of child marriage that captures recent estimates as against the standard indicator of proportion of 20-24 year-old women married by age 15/18 that has been often seen as capturing the 'incidence' of child marriage in the past. I would like to flag a few points to the attention of the authors.

1. Authors note that estimates of child marriage are produced for the year 2017, providing baseline estimates of child marriage at the dawn of the SDG tracking period which is a good rationale. However, I am wondering whether the authors could also provide an estimate for the year 2020 as it can help monitor the progress towards eliminating child marriage in the five years since the declaration of the SDGs. In fact, 23 of the 91 countries included in the study have survey data that cover the period between 2016-2020 at least from the DHS.

Response: Thank you for pointing this out. We have updated the analysis to include 58 new surveys that have become available since the original analysis. Currently, 98 LMICs are included in the study.

2. Authors have included 91 LMICs and a scan of DHS website suggests some important omissions, for example, South Africa, Tanzania, Guatemala and there are others too (Appendix Table 1), despite having data from DHS for the period that authors have relied for the analysis. I am wondering whether authors applied any exclusion criteria.

Response: Thank you. We have updated the analysis to include 58 new surveys that have become available since the original analysis. Microdata for South Africa, Tanzania, and Guatemala were not publicly accessible at the time of the original analysis, but are now included in the study. All surveys included in the study are based on all- women samples, and surveys based on ever-married samples are excluded.

3. Authors note that to estimate the absolute numbers of girls that were married in 2017, they have assumed that the age-specific probabilities and rates of child marriage that were observed in a given country at the time

of the latest survey remain constant until year 2017, and that this was a fairly reasonable assumption for the

majority of the 91 countries included in the study. Trend data from DHS suggest that some of the included countries may have experienced noticeable decline (and in rare cases, an increase) even during a short-span of 5-6 years. I am wondering whether the authors could have projected different scenario based different level of rate of child marriage.

Response: Thank you for pointing this out. We agree with the reviewer that recent changes will bias the estimation and the implication of our assumption is discussed in the methods and discussion sections. Our goal is to demonstrate an alternative method of estimating levels of child marriage using survival analysis approach. Future research using the survival analysis approach to assess the evolution of child marriage will be prioritized.

4. Authors have discussed the conceptual, methodological and data quality issues related to nuptiality data and authors have acknowledged this as a limitation. It would have been good if authors could reflect more on how such issues, particularly age misreporting may affect their estimated number of girls marrying before age 18;

Response: We appreciate the comment. We have included detailed discussions in both the background and discussion sections about the accuracy of the ages reported and the incompleteness of reporting, as well as their implications on the estimates.

5. Authors argue that data on age patterns of child marriage can serve as basis for assessing the implementation of existing minimum marriage age laws and advocating for such laws in countries where it is absent. While such laws can serve as a deterrent, marriage age misreporting around the legal age cannot be ignored.

Response: Thank you for pointing this out. We have added more discussions on the accuracy of the ages reported including age misreporting around the legal age in the background. 

6. Could authors also provide the estimated number of girls marrying before age 15 as an annexure?

Response: Thank you. We have added the estimated numbers of girls marrying before age 15 in the annex table. 

7. Could authors include regional estimates as well?

Response: Thank you. Our goal is to demonstrate an alternative method of estimating levels of child marriage using survival analysis approach and the comparison between the survival analysis and the direct approach is made at the national level. Future research using the survival analysis approach to generate regional estimates will be prioritized.

8. Although child marriage is more prevalent among girls than boys, could the authors have estimated the incidence of child marriage among boys too?

Response: Thank you for the suggestion. Our objective is to show an alternative technique for assessing the prevalence of child marriage among girls via the use of survival analysis. We agree with the reviewer that child marriage among boys is an underexplored area, and further research using the survival analysis method to produce estimates should be prioritized.

We look forward to hearing from you in due time regarding our submission and to respond to any further questions and comments you may have. 

Sincerely, 

Mengjia Liang

Corresponding Author

liang@unfpa.org

---

## [Decision Letter · Decision Letter 1]

27 Sep 2021

New Estimations of Child Marriage: Evidence from 98 Low- and Middle-Income Countries

PONE-D-21-16939R1

Dear Dr. Liang,

We’re pleased to inform you that your manuscript has been judged scientifically suitable for publication and will be formally accepted for publication once it meets all outstanding technical requirements.

Kind regards,

Bidhubhusan Mahapatra, Ph.D.

Academic Editor

PLOS ONE

Additional Editor Comments (optional):

Reviewers' comments:

Reviewer's Responses to Questions

**Comments to the Author**

1. If the authors have adequately addressed your comments raised in a previous round of review and you feel that this manuscript is now acceptable for publication, you may indicate that here to bypass the “Comments to the Author” section, enter your conflict of interest statement in the “Confidential to Editor” section, and submit your "Accept" recommendation.

Reviewer #1: All comments have been addressed

2. Is the manuscript technically sound, and do the data support the conclusions?

Reviewer #1: (No Response)

3. Has the statistical analysis been performed appropriately and rigorously? 

Reviewer #1: (No Response)

4. Have the authors made all data underlying the findings in their manuscript fully available?

Reviewer #1: (No Response)

5. Is the manuscript presented in an intelligible fashion and written in standard English?

Reviewer #1: (No Response)

6. Review Comments to the Author

Reviewer #1: (No Response)

7. PLOS authors have the option to publish the peer review history of their article (what does this mean?). If published, this will include your full peer review and any attached files.

Reviewer #1: **Yes: **Alissa Koski

---

## [Editor Report · Acceptance letter]

18 Oct 2021

PONE-D-21-16939R1 

New Estimations of Child Marriage: Evidence from 98 Low- and Middle-Income Countries 

Dear Dr. Liang:

I'm pleased to inform you that your manuscript has been deemed suitable for publication in PLOS ONE. Congratulations! Your manuscript is now with our production department. 

Kind regards, 

on behalf of

Dr. Bidhubhusan Mahapatra 

Academic Editor

PLOS ONE